# Structural characterisation of chromatin remodelling intermediates supports linker DNA-dependent product inhibition as a mechanism for nucleosome spacing

**Amanda L Hughes[†‡], Ramasubramanian Sundaramoorthy\*[†], Tom Owen-Hughes\***

Molecular Cell and Developmental Biology, School of Life Sciences, University of Dundee, Dundee, United Kingdom

**Abstract** Previously, we showed that *Saccharomyces cerevisiae* Chd1 chromatin remodelling enzyme associates with nucleosomes oriented towards the longer linker (Sundaramoorthy et al., 2018) (1). Here, we report a series of structures of Chd1 bound to nucleosomes during ongoing ATP-dependent repositioning. Combining these with biochemical experiments and existing literature, we propose a model in which Chd1 first associates oriented to sample putative entry DNA. In an ATP-dependent reaction, the enzyme then redistributes to the opposite side of the nucleosome, where it subsequently adopts a conformation productive for DNA translocation. Once this active complex extends the nascent exit linker to approximately 15 bp, it is sensed by the Chd1 DNA binding domain, resulting in conversion to a product-inhibited state. These observations provide a mechanistic basis for the action of a molecular ruler element in nucleosome spacing.

**\*For correspondence:**
r.z.sundaramoorthy@dundee.ac.uk (RS);
t.a.owenhughes@dundee.ac.uk (TO-H)

[†]These authors contributed equally to this work

**Present address:** [‡]Lonza Netherlands B.V., Geleen, Netherlands

**Competing interest:** The authors declare that no competing interests exist.

## Editor's evaluation

In previous studies the authors and others determined cryo-EM structures of the Chd1 remodeling enzyme where the location of the ATPase domain at SHL-2 was counter to that inferred from biochemical studies (SHL+2). Here, through a combination of compelling biochemical experiments and the convincing demonstration of additional structural states of Chd1 bound to nucleosomes, the authors provide an elegant resolution. Their data leads to a fundamental new model wherein initiation of DNA translocation occurs from action by the ATPase at SHL+2, and that once sufficient DNA has translocated out from the exit site, the DNA binding domain switches to bind the exited DNA. The results help resolve a key puzzle arising from earlier Chd1-nucleosome structures and hence will be valuable to the chromatin remodeling community.

## Introduction

Within eukaryotic cells, DNA is packaged as nucleosomes (*Luger et al., 1997*). Instead of being randomly distributed along DNA, nucleosomes are typically evenly distributed (*Widom, 1998*) and aligned to key regulatory features, such as transcriptional start sites (*Yuan et al., 2005*) and transcription factor binding sites (*Wang et al., 2012*). The action of ATP-dependent chromatin remodelling enzymes plays a key role in the establishment of nucleosome organisation in vivo at promoters and adjacent to transcription factor binding sites (*Gkikopoulos et al., 2011*; *Wiechens et al., 2016*; *Ocampo et al., 2016*; *Krietenstein et al., 2016*). Different members of the protein family have distinct roles in this process. Here, we focus on the action of Chd1. Chd1 is known to contribute to the establishment of promoter-based nucleosome organisation in yeast (*Gkikopoulos et al., 2011*;

*Ocampo et al., 2016*) and has the ability to generate evenly separated nucleosomes in vitro (*Lusser et al., 2005*; *Lieleg et al., 2015*). However, the mechanism by which even spacing is achieved is not fully understood.

Chromatin remodelling enzymes contain catalytic domains that include RecA folds, placing them within an extended family of DNA translocases, the superfamily II grouping (*Singleton et al., 2007*; *Flaus et al., 2006*). Consistent with this, chromatin remodelling enzymes are observed to translocate along DNA (*Lia et al., 2006*; *Zhang et al., 2006*; *Deindl et al., 2013*). The single-strand processing herpes virus helicase, NS3, provides a paradigm for understanding the mechanism of DNA translocation of superfamily II proteins. A series of crystal structures show the contacts between NS3 and the DNA tracking strand and indicate a ratcheting action during the ATP hydrolysis cycle (*Gu and Rice, 2010*). In this case, a combination of structures and biochemical studies provides a structural basis for 3′–5′ DNA translocation. Snf2-related chromatin remodelling enzymes also translocate on DNA with 3′–5′ directionality (*Lia et al., 2006*; *Deindl et al., 2013*; *Saha et al., 2005*; *Clapier et al., 2017*).

A major advance in understanding how remodelling enzymes reposition nucleosomes has been provided by structures obtained using cryo-electron microscopy (cryo-EM). In many cases, the ATPase domains are observed to bind nucleosomes at superhelical location +2 (SHL+2) or the symmetry-related location at SHL-2 (*Liu et al., 2017*; *Sundaramoorthy et al., 2017*; *Willhoft et al., 2018*; *Yan et al., 2019*; *Armache et al., 2019*; *He et al., 2020*). In some cases, structures have been obtained of enzymes bound to nucleosomes, including linker DNA on one side only. Where the enzyme includes additional domains capable of recognising this DNA, a favoured orientation for binding can be determined. This is the case for the relatively simple, single subunit remodelling enzyme, Chd1, which includes a DNA binding domain that binds to linker DNA extending to approximately 15 bp (*Sundaramoorthy et al., 2018*; *Sundaramoorthy et al., 2017*; *Farnung et al., 2017*; *Nodelman et al., 2022*). These structures also enable directionality of ATP-dependent DNA translocation to be inferred, making use of homology to NS3. Surprisingly, this indicates that Chd1 is oriented such that translocation would be anticipated to extend the long linker. This is counter to the known biochemical action of Chd1 which repositions nucleosomes towards the centre of DNA fragments so that linkers of approximately equal length are generated (*Stockdale et al., 2006*; *McKnight et al., 2011*). This has given rise to proposals that the cryo-EM structures represent an inhibited state (*Nodelman et al., 2016*; *Nodelman et al., 2017*).

Here, we show that Chd1 acts to reposition nucleosomes from SHL+2, rather than the SHL -2 location observed in previous cryo-EM structures. We also observe that the binding of Chd1 to nucleosomes changes in the presence of ATP, enabling contacts between the ATPase domains and SHL+2. We obtain structures of Chd1 bound to nucleosomes in the presence of ATP that can be considered snapshots of the reaction cycle. From these observations, we propose a mechanism involving transitions between Chd1 oriented to direct repositioning in both directions. When combined with the ability of the DNA binding domain to sample DNA length, this provides a means by which continuous cycles of remodelling could generate evenly spaced nucleosomes.

## Results

### Repositioning by Chd1 requires intact DNA on the exit side of the nucleosomes

To investigate whether the interactions with the entry or exit side of the nucleosome are required for Chd1-mediated repositioning, a series of two base pair gaps was introduced into a DNA fragment containing the 601 nucleosome positioning sequence. The positions of these gaps on the linear DNA fragment and how these are arranged when assembled into nucleosomes are illustrated schematically in *Figure 1A and B*, respectively. Chd1-driven repositioning of nucleosomes assembled on these DNA templates was directly compared to repositioning on intact DNA spiked into the same reaction but labelled with a different fluorescent dye. After the reaction was quenched, the positioning of nucleosomes on identical DNA fragments, with or without gaps was assessed by native gel electrophoresis (*Figure 1CD*). The introduction of gaps at all sites was observed to reduce the rate at which nucleosomes were repositioned. In three cases, repositioning was so slow that substantially less than 50% of nucleosomes were repositioned after 32 min. These sites positioned the gap from +15 to +35 on DNA sequences anticipated to interfere with Chd1 action if it were bound to the opposite SHL to that

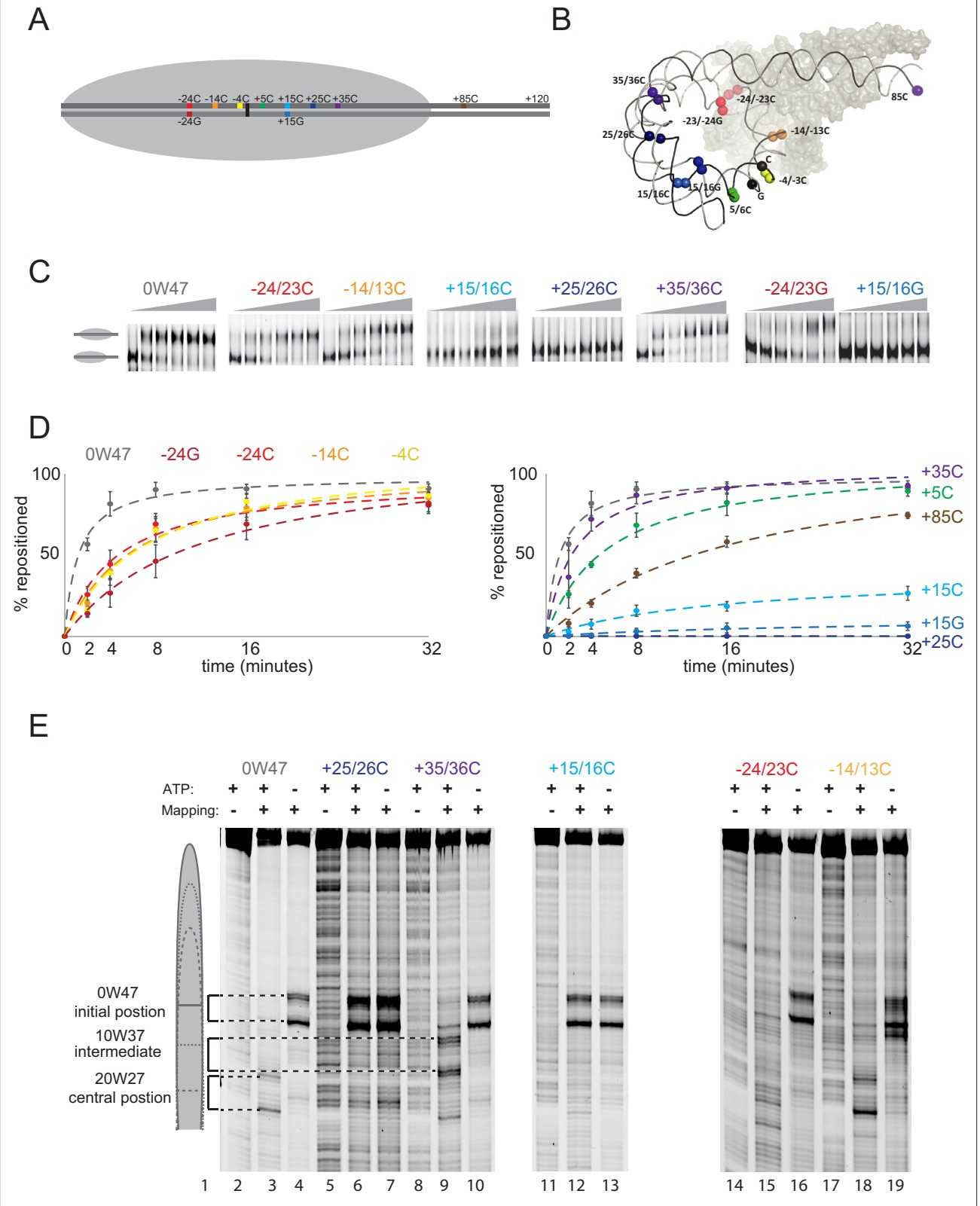

**Figure 1.** DNA gaps at SHL +2, but not at SHL -2 prevent repositioning by Chd1. (**A**) 2 bp gaps were introduced as illustrated on a linear representation of a DNA fragment, including the 601 nucleosome positioning sequence and an asymmetric 49 bp linker and (**B**) on a 3D representation of the same DNA bound by Chd1 (based on PDB 6 FTX). Positions are numbers relative to the dyad with the positive numbers proximal to extranucleosomal DNA; C or G designates the strand at which the dyad base is a C or G, respectively. (**C**) Representative native polyacrylamide gels showing how the

*Figure 1 continued on next page*

*Figure 1 continued*

repositioning of nucleosomes is affected on templates with DNA gaps introduced at the locations indicated. Repositioning reactions included 100 nM labelled nucleosomes, 5 nM Chd1, and 1 mM ATP; 10 μL aliquots were stopped with addition of 0.1 μg/ μL competitor DNA, 0.2 M sodium chloride, and 1.6% sucrose on ice immediately (t=0) and after sliding for 2, 4, 8, 16, and 32 min at 30°C. (**D**) Quantitation of 3 repeats for each DNA template as compared to intact nucleosomal DNA are fitted to a hyperbolic curve. Error bars indicate the Standard Error. (**E**) The extent of nucleosome repositioning is assessed at high resolution using site-directed hydroxyl radical cleavage directed via attachment of FeEDTA at S47C. Lane 1 contains a molecular weight marker, Lanes 2–19 include Chd1, with or without ATP as indicated and were subject to directed hydroxyl radical mapping as indicated. Nucleosomal DNA was either intact (lanes 2–4) or included 2 bp gaps as the locations indicated (lanes 5–19). Cleavage sites are detected following denaturing PAGE. Cleavages corresponding to the initial, intermediate and product nucleosome locations are indicated in the schematic to the left.

The online version of this article includes the following source data for figure 1:

**Source data 1.** Raw data files for gel scans in .tif and .adf formats displayed in *Figure 1*.

**Source data 2.** Data images in *Figure 1* as .pdf files with relevant areas labelled.

observed in cryo-EM structures in the presence of ADP-BeF$_3$ (*Sundaramoorthy et al., 2018*; *Farnung et al., 2017*; *Nodelman et al., 2022*). As this is the side of the nucleosome where new exit DNA is generated following Chd1 action, we refer to it as the exit side of the nucleosome. Gaps both on the top strand, in which the dyad base is deoxycytosine, and the bottom strand, in which it is a deoxyguanosine, had strong effects. This is consistent with previous data showing that within nucleosomes, translocation of RSC is impeded by gaps on either strand (*Saha et al., 2005*). The basis for this is now clearer as ATPase domains make extensive contact with both DNA strands (*Sundaramoorthy et al., 2018*; *Farnung et al., 2017*).

As native gel electrophoresis does not resolve small or symmetry-related changes in nucleosome positioning, histone DNA contacts were also characterised with base pair precision through coupling of iron EDTA to histone H4 at S47 (*Flaus et al., 1996*). This can be used to direct cleavage of DNA from the two copies of H4 4 bp on either side of the nucleosome dyad (*Figure 1E* lane 4). Following the repositioning of intact nucleosomes, the major product is repositioned by 20 base pairs to a location that is close to the centre of the DNA fragment (*Figure 1E* lane 3). In contrast, on gapped templates that greatly reduced repositioning, nucleosomes were substantially retained at their original locations (gaps at +25 C, +15 C). Repositioning of nucleosomes on the template with the gap at +35 results in repositioning of nucleosomes to an intermediate location (*Figure 1E* lane 9). In this case, gaps are drawn into the nucleosome until they reach locations equivalent to +25 and +15 that are inhibitory. This is consistent with previous observations with Chd1 (*McKnight et al., 2011*), Isw2 (*Zofall et al., 2006*), and RSC (*Saha et al., 2005*) all indicating that repositioning is impeded by gaps on the exit side of the nucleosome. It is also consistent with the previous observation that appropriately placed gaps do not prevent the initiation of repositioning but can limit the extent of repositioning (*Sabantsev et al., 2019*).

## Chd1 activity is blocked by occlusion of ATPase binding to entry side nucleosomal DNA

DNA gaps could potentially prevent the ATPase domains making contacts required to drive DNA across the nucleosome surface. Alternatively, they could increase DNA flexibility, preventing distortions to DNA from propagating across the nucleosome surface (*Li et al., 2019*). To distinguish between these explanations, steric blocks were introduced into nucleosomes with otherwise intact DNA strands. This was achieved through the coupling of streptavidin to biotinylated DNA bases. Binding of streptavidin has previously been shown to provide a means of blocking Chd1 binding to one side of nucleosomes while binding to the other side is not affected (*Nodelman et al., 2017*). This approach was adapted to determine which nucleosomal surface is required for repositioning by assembling nucleosomes onto 194 bp DNA fragments with biotin introduced on either side of the nucleosome dyad. In the absence of streptavidin, native gel electrophoresis indicates that nucleosomes on both biotin-labelled DNAs are repositioned to more central locations (*Figure 2AB* lanes 1–5). In the presence of streptavidin, both DNA and nucleosomes are super-shifted (*Figure 2AB* lanes 6–10), but this is not complete for nucleosomes labelled on the SHL+2 side (*Figure 2B* 6–11), most likely due to incomplete biotinylation. In the case of the SHL-2 labelled nucleosomes, a subtle shift in the mobility of the streptavidin-bound nucleosomes was observed following incubation with Chd1 (*Figure 2A* lanes 7–10). To confirm this shift in mobility represents a change in the histone octamer on this DNA fragment, reactions

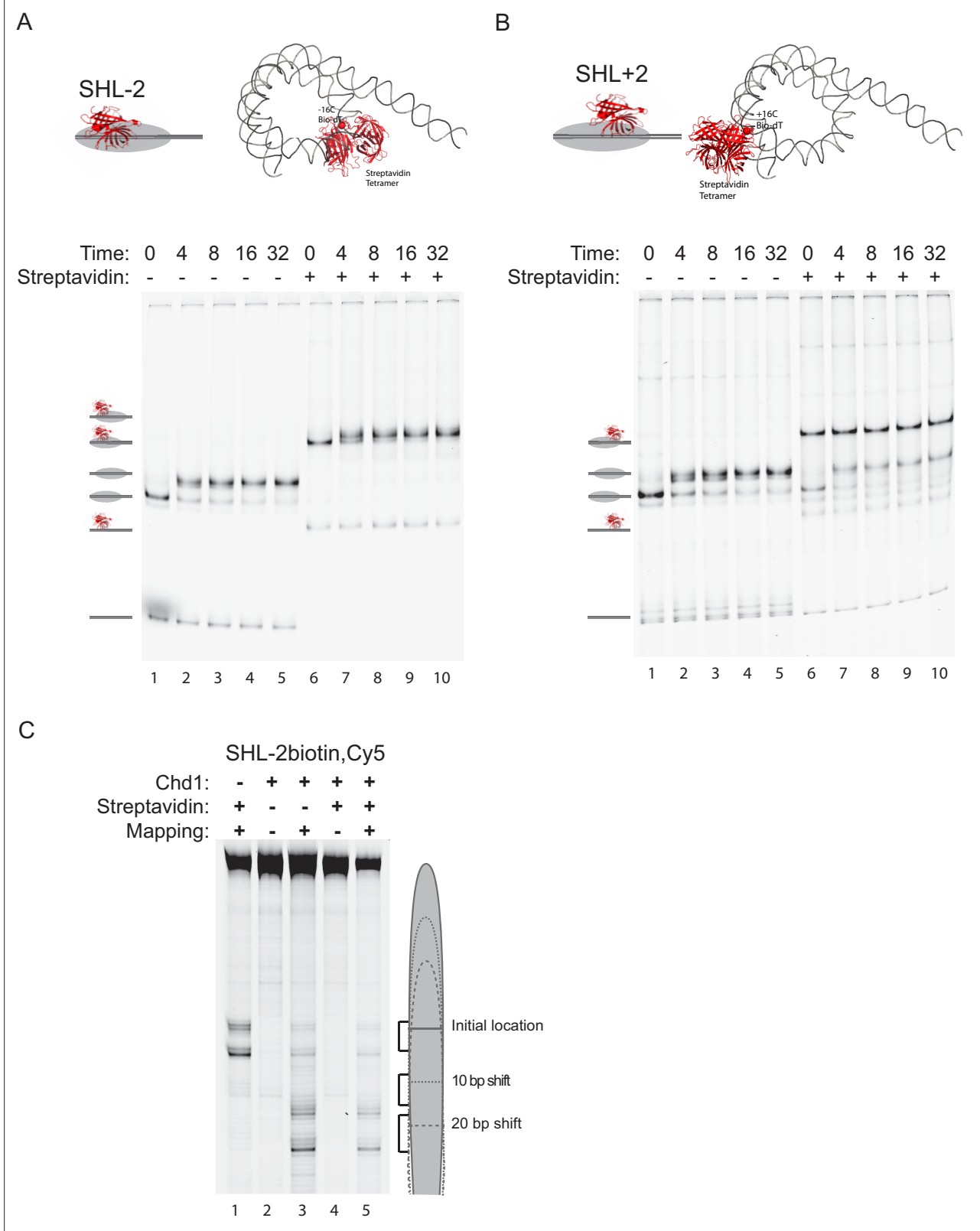

**Figure 2.** Access to SHL +2, but not SHL -2, is required for Chd1-directed nucleosome repositioning. DNA fragments in which the 601 sequence is flanked by a 49 bp linker were generated with a biotin-modified base introduced at –17 (**A**) or +17 (**B**). Nucleosomes were assembled onto these fragments and found to be repositioned to more central locations following incubation of 100 nM nucleosome with 5 nM Chd1 and 1 mM ATP. In the presence of 400 nM streptavidin, both nucleosome and DNA fragments are supershifted (Lanes 6–10). A supershift corresponding to repositioning of

*Figure 2 continued on next page*

*Figure 2 continued*

streptavidin-bound nucleosomes was only observed for nucleosomes bound by streptavidin at SHL -2. (**C**) Nucleosome repositioning was measured at high resolution using site-directed hydroxyl radical cleavage. A change in cleavage consistent with repositioning by 20 bp is observed when streptavidin is bound at SHL-2 (lanes 1–5).

The online version of this article includes the following source data for figure 2:

**Source data 1.** Raw data files for gel scans in .tif and .adf formats displayed in *Figure 2*.

**Source data 2.** Images displayed in *Figure 2* as .pdf files with relevant areas labelled.

were repeated using S47C octamers with the Fe EDTA compound attached to enable the location of histone H4 to be determined by directed DNA cleavage (*Figure 2C*). This confirms that both in the presence and absence of streptavidin, Chd1 action causes a reduction in the proportion of nucleosomes at the original location and an increase at a new location shifted 20 base pairs towards the linker DNA. A comparable shift was not observed for nucleosomes bound by streptavidin SHL +2, suggesting that binding in this region prevented repositioning.

## Chd1 binds to nucleosomes in distinct conformations in the presence of ATP

Both the introduction of DNA gaps and the steric blocking with streptavidin suggest that functional engagement of Chd1 occurs at the SHL2 location closest to entry DNA. In contrast, published structures of Chd1 all indicate that ATPase domains interact with the SHL location distal to entry DNA. To investigate whether Chd1 engages with DNA in a different conformation during ATP hydrolysis, samples were prepared in which Chd1-nucleosome complexes with a 32 bp linker DNA on one side were incubated with 100 µM ATP prior to mild cross-linking and plunge freezing onto EM grids. Following the collection of high-resolution movies, the data was processed using the CryoSPARC workflow utilising 3D variability analysis, hierarchical classification, particle subtraction, and local refinement to determine different species present within the dataset. Four major species were identified, including unbound nucleosomes and three major Chd1-bound nucleosome complexes (*Figure 3—figure supplement 1*). In complex I (*Figure 3A*), the arrangement of nucleosomal DNA is almost identical to that observed in the unbound nucleosome with both sides of the nucleosome DNA edge wrapped and well-defined DNA on the putative exit side, suggesting that repositioning has not taken place. Density corresponding to the Chd1 ATPase lobes and a portion of the chromodomain is detected at the SHL2 region closer to the short edge of the nucleosomal DNA. This contrasts with previous structures of Chd1-nucleosome complexes in which Chd1 is bound at the SHL -2 location adjacent to the longer linker DNA. We speculate that complex I represents a previously undetected state formed during activation of Chd1 in which ATPase lobes are repositioned to the opposite side of the nucleosome in an orientation that is poised for driving DNA from the longer to the short linker. As there is no evidence that DNA has been repositioned, we believe this intermediate requires additional reconfiguration prior to gaining full activity for DNA translocation.

In Complex II (*Figure 3B*), the density for linker DNA is less well defined, which may reflect a more dynamic ensemble with varying linker lengths on either side. Consistent with this, complex II has the lowest local resolution despite being most abundant by particle number (*Figure 3—figure supplement 2*). Density for entry linker DNA extends to 15 bp but is weak in comparison to the DNA associated with the histone octamer. On the exit side, DNA is partially unwrapped (*Figure 3—figure supplement 3*) but follows a trajectory intermediate between that observed in previously reported ADP-BeF$_3$ and in the apo states (*Sundaramoorthy et al., 2018*; *Sundaramoorthy et al., 2017*; *Farnung et al., 2017*; *Nodelman et al., 2022*). As in complex I, density for ATPase lobes and the chromodomain are observed. In addition, the ChEx domain (*Nodelman et al., 2022*) is associated with the acidic patch, implicating this as a key stage in activation of repositioning activity.

In Complex III (*Figure 3C*) Chd1 is bound to nucleosomes in a configuration closely related to that observed previously (*Sundaramoorthy et al., 2018*; *Sundaramoorthy et al., 2017*; *Farnung et al., 2017*; *Nodelman et al., 2022*). The DNA binding domain is bound to linker DNA, ATPase lobe 1 and the chromodomain are well resolved, whereas ATPase lobe 2 is poorly defined and the N-terminal ChEx domain is bound to the acidic patch. The DNA in this structure is not as precisely positioned as in previous structures. We observe additional density on the short linker side extending 5–7 bp further

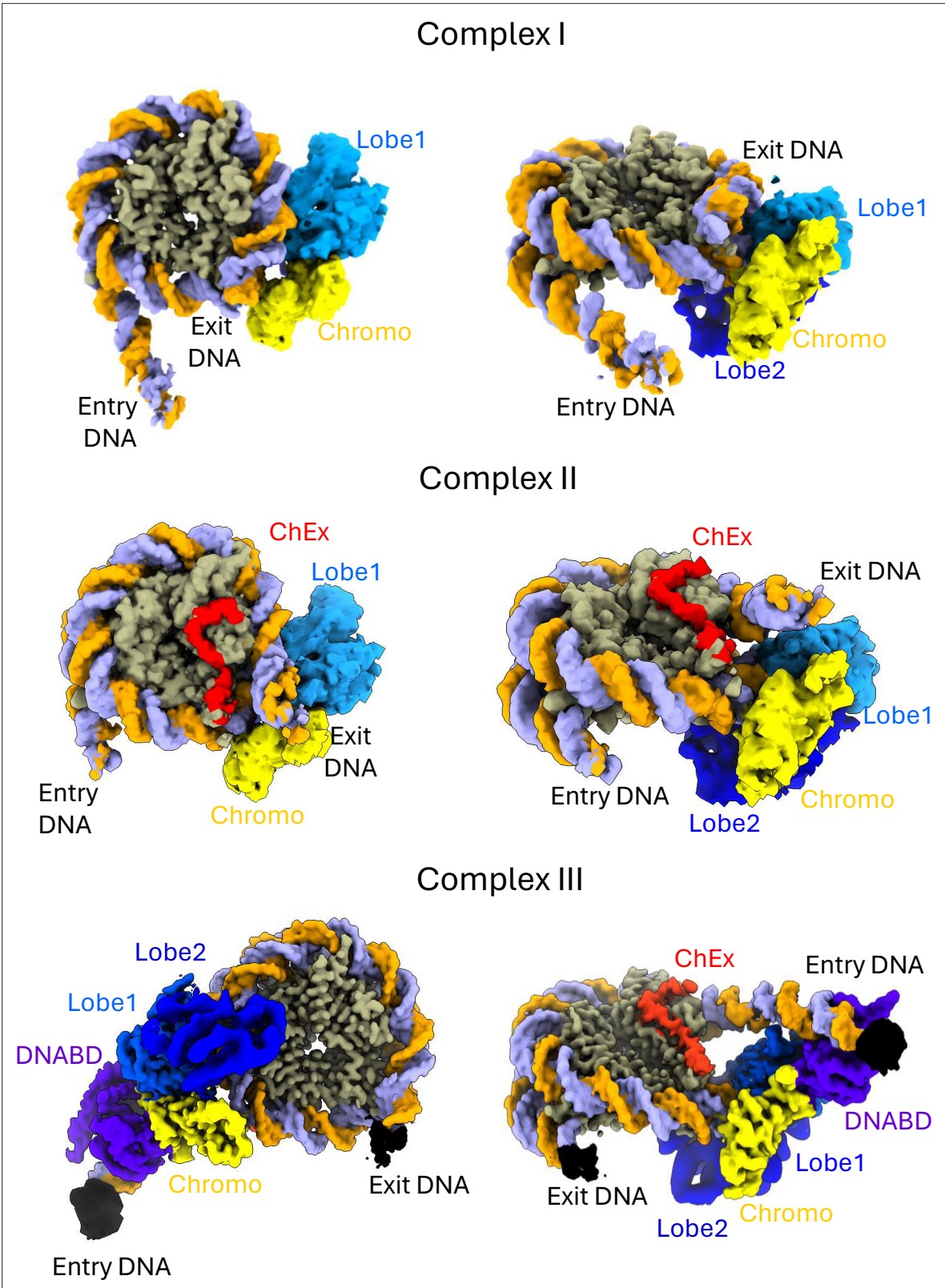

**Figure 3.** Chd1-nucleosome complexes detected in the presence of ATP. Complexes between Chd1 and nucleosomes with 29 bp linker DNA on one side were incubated with 100 µM ATP for 10 min at 25°C prior to cross-linking and cryo-grid preparation. Following data processing three major Chd1-bound complexes are detected. (**A**) Complex I with DNA fully wrapped and in an identical state to that observed in unbound nucleosomes. (**B**) Complex II with poorly defined DNA ends suggesting an ensemble of intermediates detected as a result of ongoing remodelling. (**C**) Complex III in which

*Figure 3 continued on next page*

*Figure 3 continued*

Chd1 is bound in a configuration similar to that observed in previous structures except that DNA ends are not well defined (Partial density indicated in black) suggesting some repositioning has occurred. Density is coloured as follows: DNA strands - orange and purple, histone octamer - grey, Chd1 chromodomains - yellow, Chd1 ATPase Lobe1 and Lobe2 - light and dark blue, respectively.

The online version of this article includes the following figure supplement(s) for figure 3:

**Figure supplement 1.** Image processing and 3D reconstruction of Chd1-nucleosome complexes.

**Figure supplement 2.** Local resolution and FSC resolution estimation.

**Figure supplement 3.** Distinct states of DNA wrapping in Chd1 complexes.

**Figure supplement 4.** Table illustrating key parameters for cryo-electron microscopy (cryo-EM) structures.

**Figure supplement 5.** Illustrations of fit to density map.

than observed in nucleosomes. As a result, it's likely that this structure is formed during or subsequent to ATP-dependent nucleosome repositioning.

## Chd1 ATPase domains bind to both sides of nucleosomes in the presence of ATP

To validate the ATP-dependent changes in the orientation of Chd1 engagement, FeEDTA was coupled to Chd1 at a cysteine residue engineered at 524 within ATPase lobe 1. Importantly, Chd1 proteins, including this mutation, remain active for nucleosome repositioning (*Figure 4—figure supplement 1*). DNA cleavages directed from this site show strand specificity with weak cleavage on the top (C at dyad) strand at 16/17 either in the absence of nucleotides or in the presence of ADP-BeF$_3$ (*Figure 4A* lane 2+3). On the bottom (G at dyad) strand, stronger cleavage is observed at 17/15 on the top strand (*Figure 4B* lanes 2+3). These cleavages are all consistent with the configuration of Chd1 binding previously observed in the presence of ADP-BeF$_3$ by EM and cross-linking with the ATPase domains bound to the SHL2 location distal to linker DNA (*Hauk et al., 2010*; *Sundaramoorthy et al., 2018*; *Sundaramoorthy et al., 2017*; *Farnung et al., 2017*; *Nodelman et al., 2022*). In the presence of 1.5 or 2.5 μM ATP, strong additional cleavage of the top strand and weak cleavage of the bottom strand are observed at +15 (top strand) and +18 (bottom strand). These new cleavages are consistent with repositioning of Chd1 from the familiar state in which the DNABD is bound to the entry linker and ATPase lobes bound to the adjacent SHL2 in the presence of no nucleotide or ADP-BeF$_3$ to being bound equally to both sides of the nucleosome in the presence of ATP. To reduce the possibility of nucleosome repositioning leading to altered Chd1 mapping, low concentrations (2.5 μM) of ATP were used, reactions were performed at 0°C, and nucleosomes were assembled on templates with short linkers that are known to be repositioned less efficiently (*Stockdale et al., 2006*). Mapping the locations of nucleosomes using FeEDTA coupled at H4S47C confirmed that nucleosomes are substantially retained at their initial locations under these conditions, which are slightly different to those used for cryo EM. This indicates that the binding of Chd1 to both sides of nucleosomes in the presence of ATP is likely to occur prior to nucleosome repositioning. Due to the symmetry of the nucleosome and as our structures do not resolve DNA bases, it is not possible to assign the orientation of complexes from the density maps alone. The changes in directed DNA cleavage indicate a proportion of complexes are formed on the opposite side of the nucleosome in the presence of ATP. The concentration of ATP in the reactions is below the Km for ATP, and this may limit the extent of the change in binding orientation.

## Discussion

By mapping the interactions of Chd1 with nucleosomes in solution, we have shown that Chd1 switches from binding close to entry DNA at SHL –2 in the apo and ADP-BeF$_3$ bound states to being bound on both nucleosome surfaces in the presence of ATP. The interaction of Chd1 with SHL +2 is required for repositioning, indicating that this nucleotide-dependent switch in binding orientation is required for productive nucleosome repositioning. These observations are consistent with the orientation in which Chd1 repositions asymmetrically blocked nucleosomes (*Nodelman et al., 2021*). Recent studies provide evidence for nucleotide-dependent changes to remodelling enzyme structure on different scales. Localised changes to the DNA contacts made by the ATPase domains take place as they

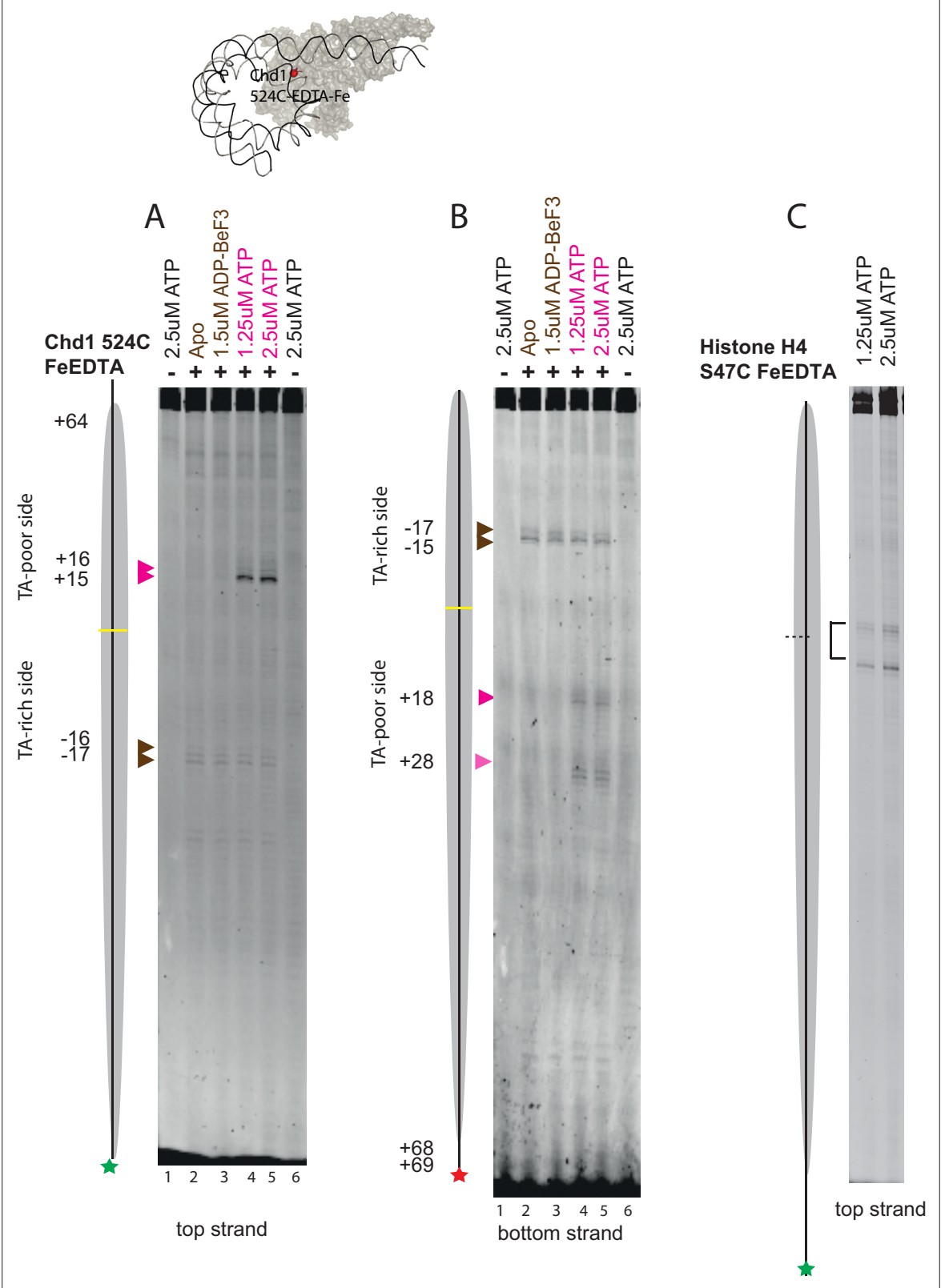

**Figure 4.** ATP-dependent changes to the binding of Chd1 ATPase domains. Hydroxyl radical mapping, directed by Chd1 S524C (300 nM), on nucleosomes (100 nM) with a 13 bp linker reports on the location of the ATPase lobes in apo, ADP-BeF$_3$-, and ATP-bound states. (**A**) shows cleavage sites detected on the top strand whereas (**B**) shows cleavage detected on the bottom strand. In the apo and ADP-BeF$_3$ states, cleavage occurs on the –2 side of the nucleosome, whereas in the presence of ATP, contact with both the +2 and –2 sides of the nucleosome is observed. Nucleosome mapping

*Figure 4 continued on next page*

*Figure 4 continued*

via hydroxyl radical cleavage shows that nucleosomes are not repositioned under the conditions of these reactions (**C**). The coloured asterisks indicate the locations of the fluorescent label used to detect DNA.

The online version of this article includes the following source data and figure supplement(s) for figure 4:

**Source data 1.** Raw data files for gel scans in .tif and .adf formats displayed in *Figure 4*.

**Source data 2.** Images displayed in *Figure 4* as .pdf files with relevant areas labelled.

**Figure supplement 1—source data 1.** Raw data files for gel scans in .tif and .adf formats displayed in *Figure 4—figure supplement 1*.

**Figure supplement 1—source data 2.** Images displayed in *Figure 4—figure supplement 1* as .pdf files with relevant areas labelled.

**Figure supplement 1.** Chd1 S524C retains nucleosome repositioning activity.

(**A**) Repositioning assays performed with 300 nM Chd1 S524C and 100 nM nucleosomes with ATP concentration and times as indicated. (**B**) Native gel showing integrity of nucleosomes used in *Figure 4A and B*.

drive DNA distortions (*Sabantsev et al., 2019*; *Sia et al., 2025*; *Winger et al., 2018*; *Malik et al., 2025*). The changes we observe occur on a larger scale and likely occur in addition to these smaller scale changes. We suspect that larger scale reconfiguration of ATPase domain interactions may be a common feature of remodelling enzymes that act to space nucleosomes evenly when relevant accessory domains are associated with the ATPase. Indeed, larger scale changes in domain interactions have been observed in ISWI enzymes (*Gangaraju et al., 2009*; *Leonard and Narlikar, 2015*).

By combining the observations made here with previous reports, we can synthesise a likely reaction cycle for the Chd1 enzyme (*Figure 5*). Initially, Chd1 is likely to associate with nucleosomes in the configuration observed previously in the absence of nucleotide (*Nodelman et al., 2022*). Upon ATP binding, at least a proportion of Chd1 ATPase domains dissociate from SHL-2 and reassociate at SHL +2, consistent with complex I (*Figure 3*) and the appearance of site-directed DNA cleavage at SHL +2 in the presence of ATP (*Figure 4*). The mechanism driving this switch is unclear; however, translocation of Chd1 by even 1 bp from the apo state would generate strain between the DNA binding domain, the ChEx domain and the ATPase lobes due to the associated ~30° change in their rotational orientation around the DNA helix. Under this strain, the ATPase lobes may be prone to dissociate as the DNA binding domain has a higher affinity for DNA. Remaining nucleosome contacts made via both the ChEx domain and DNA-binding domain may act to reduce the probability of complete Chd1 dissociation and favour reassociation at the alternate SHL +2 location. We do not detect density corresponding to the DNA binding domain in complex I, likely the result of failure to form a stable interaction with DNA as is the case in ISWI enzymes (*Sia et al., 2025*; *Malik et al., 2025*; *Chio et al., 2024*). A recent report describing distinct nucleotide-free forms of human Chd1 indicates the ability of the DNA binding domain to bind the opposite side of the nucleosomes to the ATPase domains (*James and Farnung, 2025*). This supports the concept that Chd1 domains can independently associate with different sides of the nucleosome. However, the way in which hChd1 interacts with nucleosomes is distinct from yChd1 in that it does not unravel 10 bp of DNA in the nucleotide bound state and multiple nucleotide free states are observed. Yeast Chd1 forms a well-defined apo complex (*Nodelman et al., 2022*) and repositioning of the ATPase domains to the other side of the nucleosome requires ATP hydrolysis (*Figures 3 and 4*). Following the association of ATPase domains on the other side of the nucleosome, the DNA-binding and ChEx domains could dissociate from their original locations to complete a switch between nucleosomal surfaces. This ability to efficiently associate with different sides of the nucleosome is consistent with previous observations indicating a single Chd1 has the ability to direct repositioning of nucleosomes in different directions (*Qiu et al., 2017*). Importantly, in complex I, the DNA ends are well defined and indistinguishable from those in nucleosomes not bound by Chd1 (EMD-53596). As a result, we believe Complex I is likely to be inactive for DNA translocation.

The conversion of complex I to complex II is likely to be coupled with reassociation of the ChEx domain with the acidic patch as observed in the EM structure but could involve additional autoregulatory changes not resolved at the density obtained during ongoing ATP-hydrolysis but detected in other studies (*Nodelman et al., 2025*). In complex II, the DNA on both sides of the nucleosome has poorly defined ends, which is consistent with the ratcheting action of the ATPase domains driving DNA across the nucleosome in multiple steps (*Sabantsev et al., 2019*) that are averaged within this state. This is also consistent with our observation that integrity of SHL +2, but not SHL -2, is required

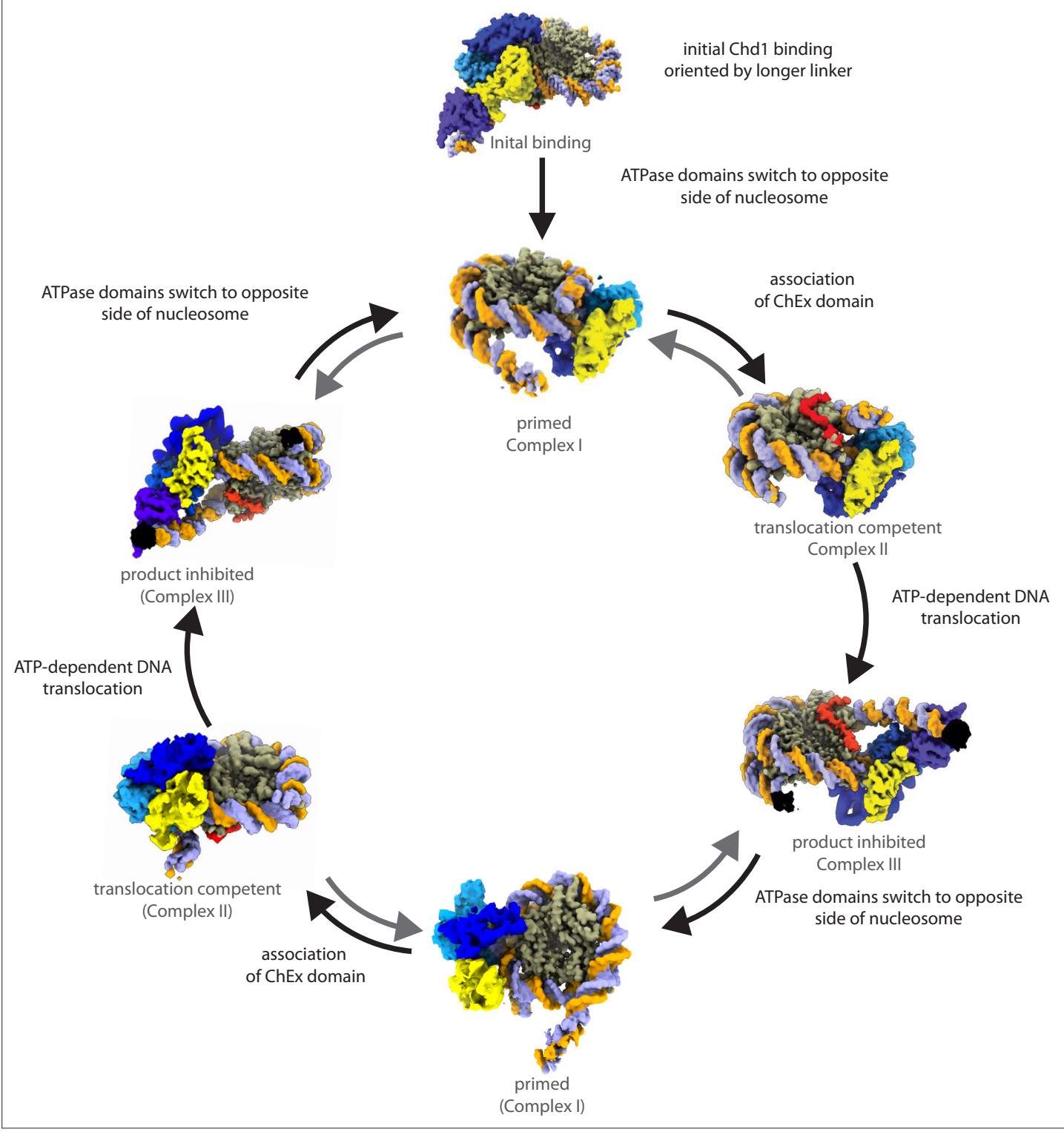

**Figure 5.** A mechanism for Chd1-mediated nucleosome spacing. Chd1 is likely to initially bind nucleosomes in the conformation previously observed in the absence of nucleotides RSC PDB 7TN2 (***Nodelman et al., 2022***). In the presence of ATP, this is converted to Complex I, in which the organisation of nucleosomal DNA has the same arrangement as nucleosomes prior to remodelling, but Chd1 is bound on the opposite side. The lack of DNA dynamics suggests Complex I is inactive for repositioning, and in the absence of linker DNA on the entry side, the location of the DNA binding domain is dynamic and so not resolved. Transition to Complex II involves association of the ChEx domain. In Complex II, the DNA ends are poorly defined, consistent with ongoing ATP-dependent translocation, which is likely to occur in single base steps. Extension of exit DNA to a length of approximately 15 bp, sufficient to bind the DNA binding domain, allows formation of Complex III. Association of the DNA binding domain is likely to constrain further DNA

*Figure 5 continued on next page*

*Figure 5 continued*

translocation, meaning this Complex III can be considered product inhibited. Via a process similar to the initial change in ATPase lobe binding, Complex III could be converted to a conformation related to Complex I, but with the ATPase lobes bound on the other side of the nucleosome. These symmetry-related complexes are labelled with brackets (Complex I). Activation of this complex would result in a second round of repositioning but with opposing directionality. Multiple cycles are anticipated to drive bidirectional repositioning tuned to favour a nucleosomal repeat length close to 15 bp. Colouring of Chd1 and nucleosomes is as in *Figure 3*.

for Chd1-mediated repositioning (*Figures 1 and 2*). In complex II, the path of exit DNA is intermediate between a fully wrapped nucleosome and one in which one turn of DNA is unwrapped as observed in previous structures (*Figure 3—figure supplement 3*). This may reflect the ability of exit DNA linkers of between 8–12 bp to associate transiently with the DNA binding domain in a conformation which partially unwraps nucleosomal DNA. In this respect, forms of complex II with increasing exit linker lengths represent intermediates in the formation of complex III.

Complex III includes approximately 15 bp of exit linker which is sufficient to enable stable association of the DNA binding and ChEx domains in the conformations observed previously (*Sundaramoorthy et al., 2018*; *Farnung et al., 2017*). We hypothesise this represents a product-inhibited state in which the association of the DNA-binding domain constrains the extension of exit DNA. This would be anticipated to reduce the rate of repositioning as exit linker length approaches 15 bp a spacing similar to that observed in vivo (*Hughes et al., 2012*). As such, this provides a molecular basis for length-dependent DNA sensing mechanism (*Eustermann et al., 2024*). The product-inhibited state is anticipated to be capable of reverting to the active state by conversion to complex I, this time located at SHL-2, enabling a subsequent round of repositioning in the opposite direction. Successive rounds of nucleosome shuffling would be anticipated to reposition nucleosomes away from barriers, such as bound transcription factors or adjacent nucleosomes (*Engeholm et al., 2024*), with a bias towards mean spacing of 15 bp similar to that observed in vivo (*Oberbeckmann et al., 2021*).

We note that the 601 DNA sequence we use to direct nucleosome assembly at specific locations is asymmetric and has been shown to influence the directionality of repositioning by Chd1 (*Winger and Bowman, 2017*). We do not believe this is likely to influence directionality in the reactions performed in this study, as these sequence-dependent effects typically bias movement on nucleosomes with two linkers. In reactions performed with on nucleosomes assembed with a single linker, as is the case in our own experiments, Chd1 repositions regardless of asymmetric sequence elements (*Levendosky et al., 2016*).

One puzzling aspect of this pathway is that Chd1 initially associates with nucleosomes in a conformation that is likely to be less active as it is closely related to the product inhibit complex III. A potential explanation for this would be that this initial interaction: (i) samples the presence of entry DNA; (ii) samples for an accessible acidic path and (iii) destabilises the entry DNA by unwrapping the outer turn. This selects for a nucleosome with entry DNA suited for subseqeunt repostioning and primes the nucleosome for more efficient repositioning following the switch to the opposite side. A related process in which Chd1 switches from acting on one side of a nucleosome to the other could be involved in the resolution of hexasome nucleosome particles (*Engeholm et al., 2024*). It is also notable that recently published structures of ISWI and SNF2H detect a series of intermediates in the formation of a nucleosome in which a single base is drawn between the ATPase domains (*Sia et al., 2025*; *Malik et al., 2025*), but not a highly dynamic state compatible with an ongoing remodelling reaction comparable to our complex II, or evidence for switching of enzymes between different sides of a nucleosome. A possible explanation for this is that accesory domains in association with the ATPase domains are required for full activity including the switching behaviour. While the single subunit enzyme Chd1 includes accessory domains in a single polypeptide, ISWI and SNF2H are likely to gain full functionality when incorporated into complexes with additional subunits that include accessory domains. We consider it likely that other ATPases that act to generate evenly spaced nucleosomes will act via pathways related to those we report for Chd1 here.

In summary, this work provides biochemical and structural evidence to support a mechanism for nucleosome spacing in which the remodelling enzyme Chd1 switches between different sides of a nucleosome to facilitate the establishment of regularly spaced arrays of nucleosomes. Structures obtained in the presence of ATP identify previously undetected intermediates supporting a mechanism in which Chd1 domains sequentially sample different sides of a nucleosome.

**Table 1.** Primers used for generating nucleosomal DNA.

| Primer | Sequence | Purpose |
|---|---|---|
| AFD 1698 | CTGCAGAAGCTTGGTCCCGGG | Unlabelled Nuc |
| AFD 1443 | Cy5-CTGCAGAAGCTTGGTCCC | Cy5 labelled Nuc |
| AFD 1629 | Cy3-CTGCAGAAGCTTGGTCCC | Cy3 labelled Nuc |
| AFD 874 | Cy5-TCAAGCTTGCATGCCTGCAGG | Cy5 labelled 49 bp (+) side extension |
| AFD 875 | Cy3-TCAAGCTTGCATGCCTGCAGG | Cy3 labelled 49 bp (+) side extension |
| AFD 1199 | Cy3-CCCCTACATGCACAGGATG | Cy3 labelled 13 bp (+) side extension |
| AFD 1203Cy5 | Cy5-CCCCTACATGCACAGGATG | Cy5 labelled 13 bp (+) side extension |
| 601C-73[49]* | CTGCAGAAGCTTGGTCCCGGGGCCGCTCAAT TGGTCGTAGCAAGCTCTA | top strand with –24 base (+) side reduction |
| 601G-73[49]* | TAGAGCTTGCTACGACCAATTGAGCGGCCCC GGGACCAAGCTTCTGCAG | bottom strand with –24 base (+) side reduction |
| 601C-22 | TCCGCTTAATCGAACGTACGCG | PCR with AFD874 to isolate top, right strand for –24/22 gap Nuc |
| 601C-73[59]* | CTGCAGAAGCTTGGTCCCGGGGCCGCTCAATT GGTCGTAGCAAGCTCTAGATCCGCTTA | top strand of Nuc with –14 base reduction on (+) side |
| 601C-12 | CGAACGTACGCGCTGTCCCCCG | PCR with AFD874 to isolate top, right strand for –14/12 gap |
| 601C-73[69]* | CTGCAGAAGCTTGGTCCCGGGGCCGCTCAATTG GTCGTAGCAAGCTCTAGATCCGCTTAATCGAACGTA | top strand of Nuc with –4 base reduction on (+) side |
| 601C-2 | CGCTGTCCCCCGCGTTTTAACC | PCR with AFD874 to isolate top, right strand for –4/2 gap |
| 601G+5 Cy5 | Cy5-GGA CAG CGC GTA CGT TCG ATT AAG | PCR with AFD1698 to isolate top, left strand for +5/7 gap and SHL-2 biotin |
| 601C+7 | CGCGTTTTAACCGCCAAGGGGA | PCR with AFD874 to isolate top, right strand +5/7 gap |
| 601G+15 Cy5 | | Cy5 labelled, PCR with 1698 to isolate top, right strand for +15/17 gap |
| 601C+17 | CCGCCAAGGGGATTACTCCCTAG | PCR with AFD874 to isolate top, left strand for +15/17 gap |
| 601C-73[99]* | CTGCAGAAGCTTGGTCCCGGGGCCGCTCAATTG GTCGTAGCAAGCTCTAGATCCGCTTAATCGAACG TACGCGCTGTCCCCCGCGTTTTAACCGCCAAG | top strand with +25 base extension on (+) side |
| 601C+27[93]* | GATTACTCCCTAGTCTCCAGGCACGTGTCAGATAT ATACATCCTGTGCATGTAGGGGATTCTCTAGAGTC GACCTGCAGGCATGCAAGCTTGA | top strand with +27 base extension on (-) side |
| 601G+35 Cy5 | /Cy5/GAG TAA TCC CCT TGG CGG TTA AAA CG | Cy5 labelled, PCR with AFD1698 to isolate top, left strand for +35/37 gap |
| 601C+37 | TAG TCT CCA GGC ACG TGT CAG AT | PCR with AFD874 to isolate top, right strand for +35/37 gap |
| 601CBio-16 (TA-rich) | CTGCAGAAGCTTGGTCCCGGGGCCGCTCAATTGG TCGTAGCAAGCTCTAGATCCGC/BiodT/TAATCGAACGTA CGCG | biotin @SHL-2, PCR with AFD874 |

*HPLC purified oligonucleotides from Eurofins.

## Methods
### Nucleosomal DNA preparation
DNA was amplified via PCR from the Widom sequence, p601.2, using the primers in *Table 1*. PCR products were concentrated via ethanol precipitation and purified by anion exchange on a Source 15Q column with a gradient from 200 mM to 2 M sodium chloride. For Chd1 mapping, PCR products were concentrated via ethanol precipitation and separated on a native 10% polyacrylamide (29:1) gel in 1X TBE at 170 V for 4 hr; DNA was electroeluted from the gel band at 90 V for 30–60 min in 1X TBE in 100 kDa cutoff Spectra/Por CE dialysis tubing. Gapped DNA was assembled from 3 pieces of

ssDNA: a full-length strand and two pieces abutting the 2 bp gap were mixed in 1:1:1 molar ratios in 10 mM Tris pH 7.5/50 mM sodium chloride and heated at 95°C for 5 min before being slowly cooled. For this purpose, PCR products were strand separated via anion exchange on a 3 mL monoQ column in 10 mM sodium hydroxide with salt gradient from 200 mM sodium chloride to 1 M sodium chloride, after a 5 min pre-treatment with 125 mM sodium hydroxide/100 mM sodium chloride. Unlabelled and Cy3-labelled ssDNA typically eluted around 76% high salt buffer, while Cy5-labelled DNA eluted over a broader region around 80–86% buffer B. For shorter segments (<100 bp) of ssDNA, HPLC-purified oligonucleotides from Eurofins were used. A similar 3-piece strategy was used for generating DNA with biotin at SHL+2, except the biotinylated ssDNA was phosphorylated before being annealed and subsequently ligated with 0W +5C ssDNA and full-length bottom strand. SHL-2 biotinylated DNA was generated via direct PCR with 601CBio-16 and AFD 874 and purified via anion exchange.

## 601.2 core sequence

601.2 core sequence:

GCGTCAGCGGGTGTTGGCGGGTGTCGGGGCTGGCTTAACTATGCGGCATCAGAGCAGATT GTACTGAGAGTGCACCATATGCGGTGTGAAATACCGCACAGATGCGTAAGGAGAAAATACCGCA TCAGGCGCCATTCGCCATTCAGGCTGCGCAACTGTTGGGAAGGGCGATCGGTGCGGGCCTCTTC GCTATTACGCCAGCTGGCGAAAGGGGGATGTGCTGCAAGGCGATTAAGTTGGGTAACGCCAGGG TTTTCCCAGTCACGACGTTGTAAAACGACGGCCAGTGAATTGTAATACGACTCACTATAGGGCG AATTCGAGCTCGGTACCCGGACCCTATACGCGGGCGCA**CTGCAGAAGCTTGGTCCCGGGGCC GCTCAATTGGTCGTAGCAAGCTCTAGATCCGCTTAATCGAACGTACGCG**cTGTCCCCCGCGTTTT AACCGCCAAGGGGATTACTCCCTAGTCTCCAGGCACGTGTCAGATATATACATCCTGTGCATGT AGGGGATTCTCTAGAGTCGACCTGCAGGCATGCAAGCTTGAGTATTCTATAGTCACC

Nucleosome positioning sequence is in bold with dyad nucleotide in lower case.

## Recombinant octamer assembly

*Xenopus laevis* recombinant histones were purified from exclusion bodies as previously described (*Luger et al., 1997*) except that cation exchange chromatography using SP Sepharose (GE) with a gradient from 0.2M to 1M sodium chloride in denaturing conditions (7 M Urea) was used for histone H2A, H2B, and H3, rather than denaturing size exclusion chromatography, as for H4 histone. For nucleosome position mapping, histone H3 C110A and histone H4 S47C mutants were used. Octamers were refolded by combining denatured (6 M guanidine hydrochloride) histones in equimolar ratios and dialysing in 3 changes of 2 M sodium chloride/10 mM Tris pH 7.5/1 mM EDTA and isolated via size exclusion chromatography (Superdex 200).

## Nucleosome assembly

Nucleosomes were assembled via salt gradient dialysis of approximately 2.5 μM *Xenopus laevis* octamer and 1.75 μM dsDNA. An acid-washed block was used for dialysis to avoid metal contamination interfering with hydroxyl radical mapping reactions. Salt was reduced stepwise in 2 hr intervals from 0.85 M to 0.65 M to 0.5 M potassium chloride in 20 mM Tris pH 7.5/1 mM EDTA; the two final dialyses were performed in 50 mM Tris pH 7.5, with the final step taking place overnight.

## Chd1 purification

Chd1 was expressed at 20°C over 48 hr in autoinduction media from a pGEX6p1 vector in Rosetta2 DE3 pLysS cells. Cells were resuspended in 40 mM Tris pH 7.5/400 mM sodium chloride with protease inhibitors and DNase I and were lysed via freeze-thawing. Clarified lysate was purified first on HisPur Cobalt Resin (Thermo) and then on Glutathione Resin (Amintra), from which the protein was cleaved via PreScission Protease overnight. Chd1 was dialysed into 40 mM Tris pH 7.5/400 mM sodium chloride/20% glycerol for storage. For Chd1 mapping, all cysteines were mutated to serines, and cysteine residues were introduced at either S524C or S1127C for mapping ATPase lobes or DBD.

## Coupling to mapping reagent

Prior to the coupling of mapping reagent, refolded octamers or Chd1 protein were reduced with 20 mM DTT for approximately 1 hr. DTT was removed over the Superdex 200 size exclusion column in their respective buffers. Upon elution, relevant fractions were combined and concentrated and

0.01 volumes of 1 M Tris pH 7.5 (chelex-treated) was added. A 100-fold molar excess of N-[S-(2-Pyridylthio)cysteaminyl]ethylenediamine-N,N,N',N'-tetraacetic Acid (TRC, p996250) was coupled to cysteine residues overnight at 4°C. Coupling to histone H4 S47C was confirmed via MALDI-TOF mass spectrometry.

## Mapping reaction

Mapping was performed on 10 μL reactions. Solutions were made in degassed water: 40 mg of ammonium iron(II) sulfate hexahydrate was dissolved in 50 mL water, 200 mg of L-ascorbic acid was dissolved in 50 mL water, and 333 μL hydrogen peroxide was diluted in 50 mL water. Ammonium iron(II) sulfate solution was diluted to 10 μL in 490 μL water and 400 μL of the ascorbic acid solution was buffered with 100 mM 1 M Tris pH 7.5 (chelex-treated) prior to their use. Iron (3.4 μL) was added to the 10 μL reaction and incubated on ice for 15 min; 5 μL each of buffered ascorbic acid and diluted hydrogen peroxide were added and hydroxyl radical cleavage of DNA progressed on ice for 45 min. Cleaved DNA was extracted via phenol:chloroform:isoamyl alcohol (25:24:1) and ethanol precipitated with the addition of glycogen carrier overnight. Prior to Chd1 mapping, Chd1 was allowed to bind the nucleosome on ice for 15 min. S47C nucleosome mapping was performed upon transfer of sliding reactions from 30°C to ice. Precipitated DNA was resuspended in 5 μL of formamide loading dye, and all of this was run for 1 hr at 50°C at 50 W maximum on a denaturing (7 M urea) polyacrylamide (8%, 19:1) gel in 1X TBE; the gel had been prerun for an hr to warm it. Cy3 and Cy5 channels were scanned on an Amersham Typhoon Gel Imaging System (GE).

## Nucleosome sliding

Nucleosome sliding was performed on 100 nM nucleosomes in 10 μL reactions in 40 mM Tris pH 7.5/50 mM potassium chloride/3 mM magnesium chloride; ATP and Chd1 concentrations are specified in the figure legends. For sliding performed on streptavidin-blocked nucleosomes, 800 nM streptavidin was preincubated for 10 min with nucleosomes in reaction buffer on ice prior to the addition of ATP and Chd1 enzyme. Reactions were stopped at the specified times by moving from 30°C to ice and adding 0.1 μg/ μL competitor DNA, 0.2 M sodium chloride, and 1.6% sucrose for loading. Nucleosomes were loaded on a prerun 6% polyacrylamide (39:1) native gel in 0.2X TBE at 300 V for 4 hr. Cy3/Cy5 labelled nucleosomes were visualised on an Amersham Typhoon Gel Imaging System (GE), and the percent of nucleosome repositioned from the DNA edge (higher mobility) towards the centre were quantitated using AIDA Image Analyzer as (intensity of higher mobility band)/(total intensities of higher and lower mobility bands)×100%. Data were fit to a hyperbola using Sigma Plot Dynamic Curve Fit.

## Sample preparation for CryoEM and data acquisition

C-terminal truncated *Saccharomyces cerevisiae* Chd1 protein ScChd1(1–1305 ΔC) and the nucleosome that contains a 30 bp linker DNA on a 601.2 Widom sequence are produced as described previously (*Sundaramoorthy et al., 2018*). ScChd1(1–1305 ΔC) nucleosome complexes formation was determined by titration and native PAGE analysis. The formed complex was then size exclusion gel filtrated using a PC 3.2/30 Superdex 200 analytical column in 20 mM Hepes pH 7.5, 120 mM NaCl, 1.5 mM MgCl$_2$. In a typical run, 50 μL s of 20 μM of sample was injected using a Dionex autoloader. 50 μL fractions were collected and analysed in Native PAGE gel, and fractions containing ScChd1-nucleosome complexes were pooled together. The complex was mildly concentrated with an Amicon ultra-0.5 100 kDa concentrator to produce a final concentration of 4 μM complex. To initiate remodelling, 100 μM of ATP was added to the complex and incubated at 4°C for 10 min then subjected to crosslinking with 0.05% final concentration of glutaraldehyde on ice for 10 min. The crosslinking was quenched with 10 μL of 1 M Tris and 10 μL of 0.1 M lysine. CryoEM grids were then prepared immediately using a vitrobot FEI Mark IV cryo plunger maintained at 4°C and 100% humidity. A 4 μl drop of sample was then applied to C-flat Holey carbon foil (400 mesh R1.2/1.3 μM) pre-cleaned with glow discharge (Quorum technologies). After a 15 s wait time, grids were double-side blotted for 4 s and plunge-frozen into −172°C liquefied ethane. Standard vitrobot filter paper Ø 55/20 mm, Grade 595 was used for blotting.

The prepared grids were initially checked for their ice quality and the particle distribution using an in-house JEOL 2200 microscope with side-entry cryo-holder operated at 200 keV and equipped with

a Gatan 4k × 4k CCD camera. Cryo-grids were then stored in liquid nitrogen and dry-shipped to the electron Bio-Imaging Centre (eBIC) located in Diamond Light Source, Oxford, UK. Data was acquired on a FEI Titan Krios transmission electron microscope (TEM) operated at 300 keV, equipped with a K3 direct electron detector (Gatan). Automated data acquisition was carried out using FEI EPU software at a nominal magnification of 81,000x. *Figure 3—figure supplement 4* summarises all the parameters used for the data collection.

## CryoEM data acquisition and analysis

Structural reconstruction of nucleosome-chd1 complex in the presence of ATP using single particle analysis was carried out with CryoSPARC suite (v4.5.3). Movies were imported into CryoSPARC, and using patch motion correction frames are aligned, and beam-induced motions correction and electron dose weighting were performed. CTF estimation using CTFFIND4 was carried out, and those micrographs with CTF fit worse than 5 Å were discarded, which resulted in 2526 images. Particles were initially picked and optimised on a subset of 200 images using Blob picker with particle diameter of 200 Å which is subsequently extended to all the images. A total of 2.7 M particles were picked and extracted with a box size of 560 pixels. The extracted particles were then inspected, filtered for any ice contamination and artifacts, and subjected to three rounds of 2D classification. The best 2D classes were selected, resulting in 1.04 M particles. At this stage, local motion correction was carried out on the 2D classification polished set of particles before being subjected to ab initio reconstruction. During ab initio reconstruction, 4 classes were generated and subjected to heterogeneous refinement. A set of 500,179 particles that are attributed to nucleosome are selected and subjected to homogenous refinement, which yielded a 3.7 Å. Subsequently, 3D variability analysis, hierarchical 3D classification, particle subtraction along with homogenous, non-uniform and local refinement were carried out to determine structures of different complexes. A complete CryoSPARC workflow of structure reconstruction of different complexes is shown in *Figure 4—figure supplement 1*. Local resolution maps are computed within CryoSPARC.

Previously determined ADP.BeF$_3$ (6FTX) and apo (7TN2) nucleosome-chd1 structure and the 601-nucleosome structure (3LZ0) were used as a starting model. The models were then rigid body docked into the respective complexes map. DNA sequence of nucleosome DNA, Histone octamer, and Chd1 on the respective complexes were built using Coot (v0.9.8.95 EL). Subsequently, the model was refined with many rounds of model correction and building with Coot and ISOLDE (1.9). The model was then refined with Phenix (v1.20.1–4487) real-space refinement. Figures were generated in ChimeraX (v1.9) and The PyMOL Molecular Graphics System (v2.5.2, Schrödinger, LLC). The following structures have been deposited EMD-53596 (unbound nucleosomes), EMD-53590 (Complex I), EMD-53597 (Complex II), and EMD-53595 (Complex III).

## Acknowledgements

This work was supported by MRC grant MR/S021647/1 and Wellcome Trust 097945. ALH was funded by an EMBO long-term fellowship ALTF 380–2015 co-funded by the European Commission (LTFCO-FUND2013, GA-2013–609409). The University of Dundee Cryo-EM facility was supported by Wellcome Trust Grant 223816/Z/21/Z, an MRC capital grant, the Scottish Centre for Macromolecular Imaging, and the University of Dundee. We thank Nicola Wiechens, Matt Toman, and Jens Michaelis for valuable discussion.

# Additional information

## Funding

| Funder | Grant reference number | Author |
| --- | --- | --- |
| Medical Research Council | MR/S021647/1 | Ramasubramanian Sundaramoorthy Tom Owen-Hughes |

| Funder | Grant reference number | Author |
|---|---|---|
| Wellcome Trust | 10.35802/097945 | Ramasubramanian Sundaramoorthy Tom Owen-Hughes |
| European Molecular Biology Organization | ALTF 380-2015 | Amanda L Hughes |
| Wellcome Trust | 10.35802/223816 | Tom Owen-Hughes |

The funders had no role in study design, data collection and interpretation, or the decision to submit the work for publication. For the purpose of Open Access, the authors have applied a CC BY public copyright license to any Author Accepted Manuscript version arising from this submission.

## Author contributions

Amanda L Hughes, Conceptualization, Formal analysis, Investigation, Methodology, Writing – original draft, Project administration, Writing – review and editing; Ramasubramanian Sundaramoorthy, Conceptualization, Data curation, Formal analysis, Funding acquisition, Investigation, Methodology, Writing – original draft, Writing – review and editing; Tom Owen-Hughes, Conceptualization, Resources, Funding acquisition, Investigation, Writing – original draft, Project administration, Writing – review and editing

## Author ORCIDs

Ramasubramanian Sundaramoorthy https://orcid.org/0000-0003-4895-0980
Tom Owen-Hughes https://orcid.org/0000-0002-0618-8185

## Decision letter and Author response

Decision letter https://doi.org/10.7554/eLife.52513.sa1
Author response https://doi.org/10.7554/eLife.52513.sa2

# Additional files

## Supplementary files
MDAR checklist

## Data availability

Cryo-EM density maps have been deposited in the EM Data Resource under accession codes EMD-53596 (nucleosome), EMD-53590 (Chd1-nucleosome complex I), EMD-53597 (Chd1-nucleosome complex II), EMD-53595 (Chd1-nucleosome complex III). The atomic coordinates have been deposited in the Protein Data Bank under accession codes PDB 9R5W (Nucleosome), PDB 9R5K (Chd1-complex I), and PDB 9R5S (Chd1-complex III). All other data generated or analysed during this study are included in the manuscript and supporting files.

The following datasets were generated:

| Author(s) | Year | Dataset title | Dataset URL | Database and Identifier |
|---|---|---|---|---|
| Sundaramoorthy R, Hughes A, Owen-hughes TA | 2025 | Structural characterisation of chromatin remodelling intermediates supports linker DNA dependent product inhibition as a mechanism for nucleosome spacing | https://www. emdataresource.org/ EMD-53596 | EMDataResource, EMD-53596 |

*Continued on next page*

*Continued*

| Author(s) | Year | Dataset title | Dataset URL | Database and Identifier |
|---|---|---|---|---|
| Sundaramoorthy R, Hughes A, Owen-hughes TA | 2025 | Structural characterisation of chromatin remodelling intermediates supports linker DNA dependent product inhibition as a mechanism of nucleosome spacing, Chd1-Nucleosome complex I. | https://www.emdataresource.org/EMD-53590 | EMDataResource, EMD-53590 |
| Sundaramoorthy R, Hughes A, Owen-hughes TA | 2025 | Structural characterisation of chromatin remodelling intermediates supports linker DNA dependent product inhibition as a mechanism of nucleosome spacing. Chd1-nucleosome complex II. | https://www.emdataresource.org/EMD-53597 | EMDataResource, EMD-53597 |
| Sundaramoorthy R, Hughes A, Owen-hughes TA | 2025 | Structural characterisation of chromatin remodelling intermediates supports linker DNA dependent product inhibition as a mechanism for nucleosome spacing | https://www.emdataresource.org/EMD-53595 | EMDataResource, EMD-53595 |
| Sundaramoorthy R, Hughes A, Owen-hughes TA | 2025 | Structural characterisation of chromatin remodelling intermediates supports linker DNA dependent product inhibition as a mechanism for nucleosome spacing | https://doi.org/10.2210/pdb9R5W/pdb | Worldwide Protein Data Bank, 10.2210/pdb9R5W/pdb |
| Gabriel F, Loew C | 2024 | Structural characterisation of chromatin remodelling intermediates supports linker DNA dependent product inhibition as a mechanism of nucleosome spacing | https://doi.org/10.2210/pdb9R5K/pdb | Worldwide Protein Data Bank, 10.2210/pdb9R5K/pdb |
| Sundaramoorthy R, Hughes A, Owen-hughes TA | 2026 | Structural characterisation of chromatin remodelling intermediates supports linker DNA dependent product inhibition as a mechanism for nucleosome spacing | https://doi.org/10.2210/pdb9R5S/pdb | Worldwide Protein Data Bank, 10.2210/pdb9R5S/pdb |

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
