## [Editor Report]

In previous studies the authors and others determined cryo-EM structures of the Chd1 remodeling enzyme where the location of the ATPase domain at SHL-2 was counter to that inferred from biochemical studies (SHL+2). Here, through a combination of compelling biochemical experiments and the convincing demonstration of additional structural states of Chd1 bound to nucleosomes, the authors provide an elegant resolution. Their data leads to a fundamental new model wherein initiation of DNA translocation occurs from action by the ATPase at SHL+2, and that once sufficient DNA has translocated out from the exit site, the DNA binding domain switches to bind the exited DNA. The results help resolve a key puzzle arising from earlier Chd1-nucleosome structures and hence will be valuable to the chromatin remodeling community.

---

## [Decision Letter]

**Decision letter after peer review:**

Thank you for submitting your article "Nucleotide dependent changes to Chd1-nucleosome interactions required to initiate repositioning" for consideration by *eLife*. Your article has been reviewed by three peer reviewers, including Geeta J Narlikar as the Reviewing Editor and Reviewer #3, and the evaluation has been overseen by a Senior Editor. and John Kuriyan as the Senior Editor. The following individuals involved in review of your submission have agreed to reveal their identity: Hitoshi Kurumizaka (Reviewer #1).

The reviewers have discussed the reviews with one another and the Reviewing Editor has drafted this decision to help you prepare a revised submission. Please aim to submit the revised version within two months. Below are the consolidated comments from all three reviewers

Summary:

CryoEM structures of the Chd1 chromatin remodeler in complex with the nucleosome provided molecular insights for how the enzyme can engage its nucleosome substrate. However, the structures from both the Cramer and Owen-Hughes labs also showed Chd1 positioned on the nucleosome in a direction opposite to what would be expected given its known nucleosome repositioning activity. With the aim of resolving this discrepancy, the authors performed several thoughtful biochemical analyses to extend their previous study of the cryo-EM structure of the yeast Chd1-nucleosome complex.

Based on their results, the authors conclude that remodeling or initiation of DNA translocation occurs from action by the ATPase at SHL+2, and that once sufficient DNA has translocated out from the exit site, the DNA binding domain switches to bind the exited DNA. This would then result in the binding arrangement observed in the EM structures. In effect, the authors suggest that the EM structures represent a stabilized product state

Overall the reviewers agreed that the types of experiments presented are essential for resolving this central puzzle that has arisen ever since the Chd1-nucleosome cryo-EM structures were published. Such resolution is therefore expected to be of much interest to the chromatin community. However, as summarized below, one concern was raised pertaining to the use of the 601 sequence. This concern needs to be experimentally addressed.

Essential Revisions:

The 601 DNA sequence was selected for high affinity binding to histone octamers and also contains asymmetrically positioned DNA elements that promote histone binding. This asymmetry in the 601 nucleosome could explain why the Chd1 ATPase domain needs to interact with SHL +2 but not SHL -2 for repositioning to occur. To discount this possibility and to more rigorously test the authors' hypothesis, the key experiments should be repeated using a different nucleosome positioning sequence. The reviewers thought that the simplest experiment would be to repeat the experiment in Figure 3 (Hydroxyl radical mapping directed by Chd1 S524C) using nucleosomes assembled on another nucleosome positioning sequence. However, the authors can of course use any of their other approaches to address this concern.

[Editors' note: further revisions were suggested prior to acceptance, as described below.]

Thank you for resubmitting your work entitled "Structural characterisation of chromatin remodelling intermediates supports linker DNA dependent product inhibition as a mechanism for nucleosome spacing" for further consideration by *eLife*. Your revised article has been evaluated by Volker Dötsch (Senior Editor) and a Reviewing Editor.

Summary:

In this revision, the authors have satisfactorily addressed the main reviewer comments and have gone further by providing new cryo-EM data, which captures states of Chd1 bound at SHL+2. The revised manuscript incorporates several changes in response to previous comments, resulting in a more robust study that makes a valuable contribution to the field of chromatin remodeling, particularly in relation to Chd1. There are some remaining issues that need to be addressed, as outlined below.

1. The authors are encouraged to incorporate into the main text a concise summary of their rebuttal explanation about why the asymmetry of the 601 sequence is not a likely reason for the observed effects.

2. If available, please provide quality control gels of nucleosomes used for remodeling

3. While the Methods section indicates that model building was performed, the resulting models are barely shown in the figures, and structural interpretations are provided without sufficient validation. Notably, the CheX domain appears to be positioned near the histone core. Because the authors suggest an interaction with the acidic patch, they should include an overlay of the map and model to demonstrate this interaction and compare it with previously reported structures. Furthermore, since the DNA adjacent to the short linker in Complex 3 plays a central role in their discussion, a figure illustrating the fit between the map and model in this region is also necessary.

4. When referencing Chd1's nucleosome spacing activity the authors should also cite Lusser et al. 2005 (PMID: 15643425) and Lieleg et al. 2015 (PMID: 25733687) as biochemical evidence.

5. The model in Figure 5 is a bit confusing. The arrows showing ATP usage in both directions should be replaced with unidirectional arrows, as ATP hydrolysis is coupled to a directional process. Reversing the process will generate ATP. We suggest converting the cycle to a linear scheme with states separated by unidirectional arrows in this order: initial apo state oriented by linker DNA -> ATPase Switched poised for remodelling -> Active remodelling -> Product Inhibited.

6. In Figure 5, since the states marked as "active" or an " poised for remodelling" were sufficiently stable to be observed by single-particle cryo-EM analysis, the authors can clarify that the structures obtained are not transient intermediates in a multi-step ATP hydrolysis pathway, but rather stabilized versions resulting from perhaps a nucleotide-free or ADP-bound state. Adding this clarification could help readers better understand the interpretation.

7. Figure 5 is not cited in the main text and should be properly referenced.

---

## [Author Response]

The reviewers have discussed the reviews with one another and the Reviewing Editor has drafted this decision to help you prepare a revised submission. Please aim to submit the revised version within two months. Below are the consolidated comments from all three reviewersSummary:CryoEM structures of the Chd1 chromatin remodeler in complex with the nucleosome provided molecular insights for how the enzyme can engage its nucleosome substrate. However, the structures from both the Cramer and Owen-Hughes labs also showed Chd1 positioned on the nucleosome in a direction opposite to what would be expected given its known nucleosome repositioning activity. With the aim of resolving this discrepancy, the authors performed several thoughtful biochemical analyses to extend their previous study of the cryo-EM structure of the yeast Chd1-nucleosome complex.Based on their results, the authors conclude that remodeling or initiation of DNA translocation occurs from action by the ATPase at SHL+2, and that once sufficient DNA has translocated out from the exit site, the DNA binding domain switches to bind the exited DNA. This would then result in the binding arrangement observed in the EM structures. In effect, the authors suggest that the EM structures represent a stabilized product stateOverall the reviewers agreed that the types of experiments presented are essential for resolving this central puzzle that has arisen ever since the Chd1-nucleosome cryo-EM structures were published. Such resolution is therefore expected to be of much interest to the chromatin community. However, as summarized below, one concern was raised pertaining to the use of the 601 sequence. This concern needs to be experimentally addressed.

We thank the reviewers for their appreciation of the significance of the work.

Essential Revisions:The 601 DNA sequence was selected for high affinity binding to histone octamers and also contains asymmetrically positioned DNA elements that promote histone binding. This asymmetry in the 601 nucleosome could explain why the Chd1 ATPase domain needs to interact with SHL +2 but not SHL -2 for repositioning to occur. To discount this possibility and to more rigorously test the authors' hypothesis, the key experiments should be repeated using a different nucleosome positioning sequence. The reviewers thought that the simplest experiment would be to repeat the experiment in Figure 3 (Hydroxyl radical mapping directed by Chd1 S524C) using nucleosomes assembled on another nucleosome positioning sequence. However, the authors can of course use any of their other approaches to address this concern.

The experiment proposed is not trivial as a strong nucleosome positioning sequence is required to obtain well defined nucleosomes that provide interpretable hydroxyl mapping data. While it is true that DNA structural properties can bias the directionality with which of ATP-dependent reposition directed Chd1 (Winger and Bowman 2017), these effects are typically observed to bias the direction of movement on nucleosomes with two linkers. Our experiments include an asymmetric linker and in these cases Chd1 repositions regardless of asymmetric DNA elements e.g. Figure 4 (Levendosky, Sabantsev et al. 2016). We also not that the structure of Chd1-nucleosome complexes are essentially indistinguishable when Chd1 is bound to either side of the 601 position sequences (compare the structures reported by Sundaramoorthy, Hughes et al. 2018; Farnung, Vos et al. 2017). As a result, we do not believe sequence bias is a plausible explanation for the effects we observe.

[Editors’ note: what follows is the authors’ response to the second round of review.]

Summary:In this revision, the authors have satisfactorily addressed the main reviewer comments and have gone further by providing new cryo-EM data, which captures states of Chd1 bound at SHL+2. The revised manuscript incorporates several changes in response to previous comments, resulting in a more robust study that makes a valuable contribution to the field of chromatin remodeling, particularly in relation to Chd1. There are some remaining issues that need to be addressed, as outlined below.1. The authors are encouraged to incorporate into the main text a concise summary of their rebuttal explanation about why the asymmetry of the 601 sequence is not a likely reason for the observed effects.

We have included a summary in the revised discussion.

2. If available, please provide quality control gels of nucleosomes used for remodeling

The integrity of nucleosomes was checked by native gel electrophoresis prior to all remodelling reactions. In Figure 1 images of gels showing the quality of samples are shown in panel C. To save space these gels were cropped. However, the full gels are included in the source data and all species present from the wells to the DNA can be seen. We think the cropping of the gels is justified as these include the key information relevant to the experiment. Similarly in Figure 2, the native gels (Figure 2A and B) indicate the quality of the reconstitutions. For the structural studies shown in Figure 3, native page was used to check the quality of nucleosomes prior to structural characterisation. Consistent with this the density for the histone octamer is strong indicating intact nucleosomes and no evidence for nucleosome lacking a subset of histones. For Figure 4, Figure supplement 1 indicates the integrity of nucleosomes during remodelling with the 524C mapping mutant. This was performed using nucleosomes assembled onto a 0W47 DNA fragment. We have added an additional panel, Figure 4 supplement 1B, indicating the quality of nucleosomes assembled on the 0W11 fragment used in Figure 4. When characterising Chd1-nucleosome complexes, native gel electrophoresis is not ideal as binding is likely to be affected by the change in ionic conditions from solution to in gel. Instead, we characterise these complexes by separately, measuring the interaction of Chd1524C (Figure 4 AB), Histone H4 (Figure 4C), and by determining structures (Figure 3).

3. While the Methods section indicates that model building was performed, the resulting models are barely shown in the figures, and structural interpretations are provided without sufficient validation. Notably, the CheX domain appears to be positioned near the histone core. Because the authors suggest an interaction with the acidic patch, they should include an overlay of the map and model to demonstrate this interaction and compare it with previously reported structures. Furthermore, since the DNA adjacent to the short linker in Complex 3 plays a central role in their discussion, a figure illustrating the fit between the map and model in this region is also necessary.

We have added this as a new supplement, Figure 3 Figure supplement 5. This shows fits to density for each complex showing fits for the octamer, DNA and ATPase lobes separately. The final panel shows the fit of the CheX domain showing that this s bound in complexes II and III in a very similar way to that observed previously in the structure 7TN2.

4. When referencing Chd1's nucleosome spacing activity the authors should also cite Lusser et al. 2005 (PMID: 15643425) and Lieleg et al. 2015 (PMID: 25733687) as biochemical evidence.

A small section of text including these references has been added to the introduction.

5. The model in Figure 5 is a bit confusing. The arrows showing ATP usage in both directions should be replaced with unidirectional arrows, as ATP hydrolysis is coupled to a directional process. Reversing the process will generate ATP. We suggest converting the cycle to a linear scheme with states separated by unidirectional arrows in this order: initial apo state oriented by linker DNA -> ATPase Switched poised for remodelling -> Active remodelling -> Product Inhibited.

We appreciate the reviewers pointing out that ATP usage is shown in both directions. The figure has been substantially revised only including ATP hydrolysis where the step is known to be ATP dependent. Many of the steps are potentially reversible and this is now indicated. The figure was retained in a circular form as this is required to enable multiple cycles of ATP-dependent repositioning with different directionalities. This represents a means by which Chd1 could evenly spacing nucleosomes as it is observed to in cells.

6. In Figure 5, since the states marked as "active" or an " poised for remodelling" were sufficiently stable to be observed by single-particle cryo-EM analysis, the authors can clarify that the structures obtained are not transient intermediates in a multi-step ATP hydrolysis pathway, but rather stabilized versions resulting from perhaps a nucleotide-free or ADP-bound state. Adding this clarification could help readers better understand the interpretation.

We cannot add further clarification here. The structures could be intermediates in a multi-step ATP dependent reaction. In fact, we favour this interpretation. The text states that Chd1 bound in the initial orientation may be destabilised by ATP-dependent DNA translocation. The structures are not of sufficient resolution to determine whether ATP or ADP is bound.

7. Figure 5 is not cited in the main text and should be properly referenced.

Figure 5 is now cited from the main text.